# ADDITIVE COUPLING OF LIQUID NEURAL NETWORKS AND MODERN HOPFIELD LAYER FOR REGRESSION

## ABSTRACT

Regression tasks on complex datasets often involve diverse feature interactions, long-range dependencies, and structured patterns that must be recalled across examples for accurate prediction. Conventional models—such as MLPs, tree ensembles, or standard continuous-time networks, struggle to maintain predictions and stability over extended horizons, especially when patterns must be reused. To address these challenges, we introduce a hybrid architecture that couples Liquid Neural Networks (LNNs) with Modern Hopfield Networks (MHNs) using additive fusion. The LNN component delivers input-adaptive continuous dynamics, while the associative memory enables retrieval and correction using previously encountered global structures. This biologically-inspired design preserves adaptability and stability, while leveraging memory-based recall for consistent predictions. On the OpenML-CTR23 regression benchmark, our approach consistently improved performance, with mean and median gains of 10.42% and 5.37%. These results demonstrate the effectiveness of integrating continuous dynamics and content-addressable memory for complex regression scenarios.

## 1 INTRODUCTION

Modern machine learning systems increasingly face the challenge of modeling tabular regression data that is heterogeneous, multi-scale, and structurally complex (Somvanshi et al., 2024). Such data arises in fields like healthcare, finance, recommendation systems, climate science, and industrial processes, where observations combine diverse feature types—continuous, categorical, relational—and exhibit dependencies spanning multiple scales (Jiang et al., 2025; Hollmann et al., 2025). Beyond local correlations, many regression problems demand capturing long-range structures such as recurring feature patterns, slow-evolving trends, and global consistency constraints (Lu et al., 2025). These requirements make tabular regression fundamentally different from pure classification tasks, whose outputs are discrete and bounded.

Traditional neural network architectures, such as multilayer perceptrons (MLPs) or convolutional models, typically assume localized receptive fields, independent feature processing, or short-range dependencies. While effective for static classification benchmarks, such inductive biases prove limiting in regression contexts where continuous-valued predictions accumulate error, requiring stability and precise recall of extended structure (Chen, 2025; Haber & Ruthotto, 2017). Regression tasks thus expose distinctive weaknesses in common architectures: outputs must be numerically accurate and consistent across long horizons, rather than merely separated by decision boundaries (Somvanshi et al., 2024).

Dynamic neural systems like Liquid Neural Networks (LNNs) (Hasani et al., 2018) address part of this challenge by introducing input-adaptive continuous dynamics that evolve states based on feature interactions. LNNs have proven effective for capturing fine-scale adaptivity and stability: their liquid neurons respond with variable sensitivity depending on input context, mimicking the adaptability of biological neurons. However, their adaptation is inherently local in time and feature space. LNNs lack mechanisms for pattern storage and reuse, which becomes particularly consequential in tabular regression tasks requiring retrieval of global structure, repeated combinations of features, or contextual corrections against slow drifts (Pawlak et al., 2024). In biological cognition, such functions are supported through associative memory systems that complement dynamic processing with structured recall (Wang & Cui, 2018).

To address this gap, we propose a hybrid architecture that augments LNNs with Modern Hopfield Networks (MHNs) via additive coupling. While the liquid encoder endows the system with adaptive continuous-state processing, the MHN provides associative memory retrieval that enables recurrence to previously observed feature patterns and reinforcement of long-range predictors Ramsauer et al. (2021). This combination allows local adaptability and global recall to interact seamlessly: retrieved memory patterns are injected directly into the liquid state, stabilizing evolution and improving predictive accuracy in regression. Unlike more complex gated controllers, the additive formulation preserves computational efficiency while benefiting from memory-based correction.

We evaluate the approach on the OpenML-CTR23 benchmark(Fischer et al., 2023), a diverse suite of heterogeneous tabular regression problems. Our findings show that coupling LNNs with MHNs consistently improves regression accuracy over both standard tabular baselines and vanilla liquid architectures. Beyond error reduction, the model exhibits better calibration and smoother optimization landscapes, highlighting that associative recall complements dynamic processing in a principled and stable manner.

Our contributions are summarized as follows:

1. We introduce a hybrid architecture that augments Liquid Neural Networks with Modern Hopfield Networks through additive coupling, uniting adaptive dynamics with associative memory.

2. We demonstrate that memory-based pattern retrieval stabilizes liquid dynamics and significantly improves predictive accuracy in heterogeneous tabular regression.

3. We present an extensive empirical study across 34 CTR23 datasets, showing consistent improvements over strong baselines in accuracy, calibration, and stability.

## 2 RELATED WORKS

### 2.1 CONTINUOUS-TIME NEURAL NETWORKS (CTNNs)

Continuous-time neural networks (Hasani et al., 2022) extend standard discrete computation into a differential framework, embedding temporal dynamics directly into the model architecture. Neural Ordinary Differential Equations (Neural ODEs) (Chen et al., 2019) first demonstrated how continuous transformations could be parameterized by neural networks, offering adaptive depth and efficiency in modeling evolving processes. Despite their advantages, Neural ODEs often face practical issues such as high solver cost, numerical instability under stiff dynamics, and degraded performance with noisy or irregular data (Murugesh et al., 2025).

Liquid Neural Networks (LNNs) emerged as an alternative that alleviates some of these limitations by introducing input-dependent time constants (Hasani et al., 2018). Each neuron dynamically adjusts its temporal sensitivity, enabling the network to capture multi-scale behaviors in a stable and bounded manner. This biologically inspired mechanism has proven effective in classification and forecasting, particularly in tasks involving heterogeneous features or varying timescales (Kumar et al., 2023). However, the adaptation in LNNs remains local: they evolve hidden states continuously but lack mechanisms for retaining or recalling structured patterns over longer horizons. This makes them less effective in regression settings where repeated structures and global consistency are central to predictive accuracy.

### 2.2 MEMORY-AUGMENTED NEURAL NETWORKS

Neural networks with external memory modules were introduced to address precisely this limitation: providing models with content-addressable recall and long-term reasoning capabilities (Sukhbaatar et al., 2015). Early architectures such as Neural Turing Machines (Graves et al., 2014) and Differentiable Neural Computers (Azarafrooz, 2022) augmented recurrent backbones with differentiable read–write operations, enabling sequence models to store and retrieve information beyond their bounded hidden states. While powerful, these designs were often complex to train and computationally expensive.

More recent approaches focus on fixed-form associative memories. Modern Hopfield Networks (MHNs) extend classical Hopfield attractor networks by enabling exponentially large storage capac-

ity and stable one-step retrieval. Through an energy minimization process, MHNs converge queries toward stored prototypes, effectively performing pattern completion and denoising. These properties make MHNs particularly well-suited for scenarios requiring recall of previously observed patterns. Although their adoption has been widespread in classification, vision reconstruction, and denoising tasks, their application to regression and tabular domains remains underexplored (Kashyap et al., 2024). In such settings, associative recall could serve as a corrective mechanism, anchoring predictions to recurring feature patterns and mitigating long-horizon drift.

### 2.3 Hybrid Architectures

Hybrid models that combine neural encoders with external memory have demonstrated advantages in language modeling, decision-making, and few-shot learning (Graves et al., 2016; Panchendrarajan & Zubiaga, 2024). In temporal domains, memory modules can mitigate the limitations of bounded context windows by allowing explicit access to historical patterns He et al. (2020). Recent efforts have explored combining CTNNs with attention mechanisms, transformers, or variational memories to enhance long-range reasoning (Chen et al., 2023).

Closer to the motivation of our paper, neuroscience-inspired models have investigated recurrent loops between cortical dynamics and hippocampal memory, showing that memory-supported feedback stabilizes temporal processing (Shimbo et al., 2025). However, most of these efforts rely on complex multi-stage training or gated controllers, which introduce additional design and optimization challenges. Our work takes a simpler approach: additive coupling between liquid dynamics and Hopfield retrieval. By directly injecting retrieved prototypes into the evolving hidden state, we capture both local adaptability and global recall without introducing heavy gating overhead. This design choice is aligned with the biological intuition that cortical dynamics are continually modulated by hippocampal recall, forming a lightweight but effective feedback loop.

## 3 Method

We present a regression framework that couples Liquid Time-Constant (LTC) networks with Modern Hopfield Networks (MHNs) through additive fusion. The LTC encoder provides input-adaptive continuous dynamics, while the MHN contributes associative recall of global patterns. The two modules complement one another: LTC ensures flexible local adaptation, and MHN provides global stability through memory correction. The architecture is illustrated in Figure 1.

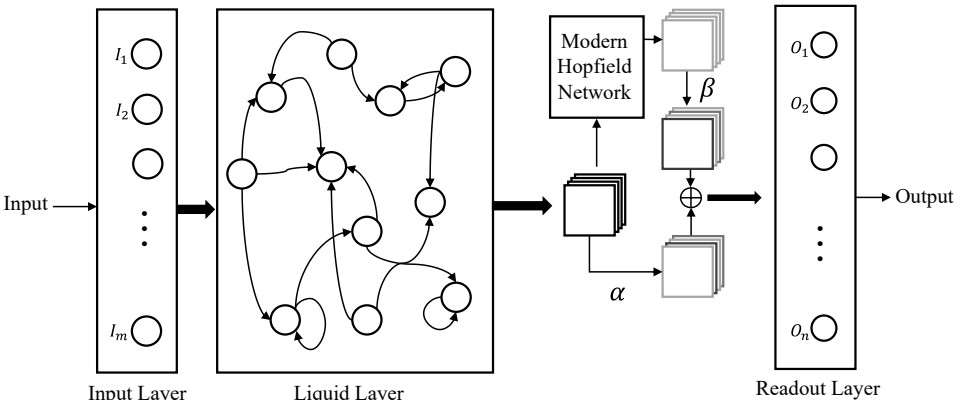

Figure 1: Schematic of the proposed hybrid architecture. Liquid dynamics encode input-dependent temporal states, which are projected into a Hopfield memory for associative retrieval. Retrieved prototypes are injected back into the liquid state via additive coupling.

## 3.1 Temporal Encoding with Liquid Time-Constant Networks

The backbone of our architecture is the Liquid Time-Constant (LTC) network (Hasani et al., 2018), a biologically inspired continuous-time model with input-adaptive dynamics. For hidden state $\mathbf{x}(t) \in \mathbb{R}^n$, the evolution is given by

$$\frac{d\mathbf{x}(t)}{dt} = -\left(\frac{1}{\boldsymbol{\tau}} + f_\theta(\mathbf{x}(t), \mathbf{I}(t))\right) \odot \mathbf{x}(t) + f_\theta(\mathbf{x}(t), \mathbf{I}(t)) \odot \mathbf{A}, \quad (1)$$

where $\boldsymbol{\tau} \in \mathbb{R}^n$ is a learnable base time constant, $\mathbf{A} \in \mathbb{R}^n$ a saturation vector, and $f_\theta(\cdot)$ a shared MLP. This yields input-dependent temporal responses, enabling neurons to react with variable sensitivity. For stable and accurate integration, we discretize Eq. 1 using a fourth-order Runge–Kutta solver.

**Lemma 1 (Boundedness of LTC states).** If $\mathbf{x}(0)$ is bounded and $f_\theta$ is Lipschitz-continuous with bounded range, then $\mathbf{x}(t)$ remains bounded for all $t \geq 0$.

*Sketch.* The system can be written as $\dot{\mathbf{x}} = g(\mathbf{x}, \mathbf{I})$, where $g$ is Lipschitz and coercive. Standard results from ODE stability theory (see Appendix A) imply forward completeness, ensuring bounded hidden states.

## 3.2 Associative Memory via Modern Hopfield Networks

LTCs effectively capture local dynamics but lack an explicit memory mechanism. We therefore integrate a Modern Hopfield Network (MHN) (Ramsauer et al., 2021), which stores a set of $N$ prototypes $\Xi = \{\boldsymbol{\xi}_1, \ldots, \boldsymbol{\xi}_N\} \subset \mathbb{R}^M$ and retrieves stored patterns given a query.

At time $t$, we compute a query from the LTC state:

$$\mathbf{q}(t) = \mathbf{W}_q \mathbf{x}(t), \quad \mathbf{W}_q \in \mathbb{R}^{M \times n}. \quad (2)$$

The MHN retrieves a prototype by soft energy minimization:

$$\mathbf{r}(t) = \sum_{i=1}^{N} \text{softmax}_i \left(\beta \cdot \mathbf{q}(t)^\top \boldsymbol{\xi}_i\right) \cdot \boldsymbol{\xi}_i, \quad (3)$$

where $\beta > 0$ is an inverse temperature controlling retrieval sharpness.

**Lemma 2 (Contraction property of MHN).** Suppose $\|\mathbf{q}(t)\| \leq R$ and $\|\boldsymbol{\xi}_i\| \leq S$ for all $i$. Then the mapping $\mathbf{q} \mapsto \mathbf{r}$ defined in Eq. 3 is Lipschitz with constant $L < 1$, making it a contraction.

*Sketch.* The retrieval can be viewed as a softmax-weighted convex combination of bounded vectors. Differentiating with respect to $\mathbf{q}$ yields Jacobian entries bounded by $\beta RS$ under softmax normalization. For sufficiently small $\beta$ or bounded $RS$, $L < 1$ holds, guaranteeing contraction. Proof details are in Appendix B.

## 3.3 Additive Coupling of Dynamics and Memory

To combine liquid dynamics with associative recall, we use a scalar-gated additive coupling:

$$\mathbf{z}(t) = \alpha \cdot \mathbf{x}(t) + \delta \cdot \mathbf{r}(t), \quad (4)$$

with $\alpha, \delta \geq 0$ as learnable scalars. This formulation balances raw liquid evolution with memory correction, while avoiding destructive interference from higher-dimensional gating matrices.

**Lemma 3 (Boundedness of coupled state).** If $\mathbf{x}(t)$ and $\mathbf{r}(t)$ are bounded, then $\mathbf{z}(t)$ is bounded for all $t$.

*Sketch.* Directly from Eq. 4, $\|\mathbf{z}(t)\| \leq \alpha\|\mathbf{x}(t)\| + \delta\|\mathbf{r}(t)\|$. Since both terms are bounded, $\mathbf{z}(t)$ is bounded.

**Lemma 4 (Gradient smoothing).** Let $\mathcal{L}$ be a differentiable loss. Then the gradient through $\mathbf{z}(t)$ decomposes as

$$\nabla_{\mathbf{z}}\mathcal{L} = \alpha\nabla_{\mathbf{x}}\mathcal{L} + \delta\nabla_{\mathbf{r}}\mathcal{L}.$$

Thus, coupling acts as a convex combination of gradient flows, reducing variance and aiding convergence. It is explained in detail in Appendix A.4.

## 3.4 Regression Head

The fused representation $\mathbf{z}(t)$ is passed to a lightweight regression head:

$$\hat{y}(t) = \mathrm{MLP}_{\mathrm{reg}}(\mathbf{z}(t)),$$

shared across timesteps and optimized end-to-end with mean squared error loss.

**Proposition (Stability of the coupled system).** By Lemmas 1–3, the coupled system admits bounded hidden states under bounded inputs. By Lemma 2, retrieval is contractive, and by Lemma 4, gradients are smoothed. Together, these ensure the coupled architecture yields stable forward dynamics and more regular optimization landscapes. A complete proof is provided in Appendix.

## 4 Experimentation and Results

This section evaluates the proposed model on the CTR23 regression suite, against strong tabular baselines. We report test performance using RMSE (primary), MAE, and $R^2$. Beyond point metrics, we analyze calibration via parity plots and training stability via 3D loss–landscape visualizations.

### 4.1 Datasets and Setup

**Datasets.** OpenML Curated Tabular Regression benchmarking suite 2023(OpenML-CTR23), a collection of 34 regression problems that meet a large number of quality criteria. It follows many of the design choices of the OpenML-CC18 (Fischer et al., 2023), which is the first benchmarking suite for classification algorithms that was created using rigorous inclusion criteria, and CTR23 was then refined for regression. CTR23 spans heterogeneous regression problems in housing, energy, materials, economics, and simulation. Each dataset comes with a prescribed train/test split. The reported results are average across 5 cross-validation sets.

**Preprocessing.** We apply simple, reproducible tabular preprocessing: (i) median imputation for numeric features, (ii) most–frequent imputation for categorical features, (iii) standardization of numeric columns, and (iv) one–hot encoding for categorical columns. All transformations are fit on the training split only.

**Models and training.** All neural models are implemented in PyTorch and trained on a single NVIDIA RTX A6000. Optimizer is Adam, loss is MSE, batch size is 256, and we use a 10% validation split for early stopping. The LTC encoder is discretized with a 4th–order Runge–Kutta solver. Learning rate was set to $0.001$, Hopfield size was set to 16, scaling-factor $\beta$ was set to $0.25$, and number of heads was set to $4$.

**Baselines.** We compare against *XGBoost* (Chen & Guestrin, 2016), *Random Forest* (Louppe, 2015), *Generalized Additive Models* (GAM) (Zhuang et al., 2020), *Ridge Regression* (Dabo & Bigot, 2025), and a *Regression Tree* (Zhang et al., 2023), alongside the *vanilla LTC* encoder. Hyperparameters follow common practice for CTR evaluations.

### 4.2 Results

Across CTR23, LTC outperforms classical tabular regressors on a majority of datasets (Table 1).Representative gains include Concrete, California Housing, and Kin8nm. These trends hold in per-metric comparisons (Table 4), where LTC yields lower RMSE/MAE and higher $R^2$ than non-continuous baselines.

Building on this, our proposed model further reduces error on 29/34 tasks, with mean and median relative RMSE gains of 10.42% and 5.37%, respectively over LTC (Table 1, ablation-Table4). The

| DATASET | XGBOOST | RF | GAM | RIDGE | TREE | LTC | PROPOSED | $\times$ |
|---|---|---|---|---|---|---|---|---|
| Abalone | 2.118 | 2.133 | 2.120 | 2.330 | 2.404 | 2.117 | **2.108** | $10^0$ |
| Airfoil Self Noise | 1.170 | 2.203 | 4.588 | 4.930 | 4.414 | 1.167 | **1.139** | $10^0$ |
| Brazilian Houses | 0.446 | 0.587 | 0.321 | 0.442 | 1.149 | 0.283 | **0.272** | $10^4$ |
| California Housing | 4.464 | 5.050 | 6.193 | 7.247 | 7.809 | 4.082 | **3.799** | $10^4$ |
| Cars | 2.111 | 2.486 | 2.935 | 3.080 | 3.422 | 2.108 | **2.095** | $10^3$ |
| Concrete Compressive Strength | 0.371 | 0.529 | 0.963 | 1.075 | 0.900 | 0.286 | **0.154** | $10^1$ |
| CPS88 Wages | 3.800 | 3.830 | 3.856 | 4.120 | 4.027 | 3.464 | **3.257** | $10^2$ |
| CPU Activity | 2.190 | 2.461 | 2.714 | 9.984 | 4.767 | 2.106 | **2.082** | $10^0$ |
| Diamonds | 0.521 | 0.540 | 1.272 | 1.335 | 1.311 | 0.495 | **0.446** | $10^3$ |
| Energy Efficiency | 0.280 | 1.082 | 2.934 | 3.298 | 2.575 | 0.277 | **0.192** | $10^0$ |
| FIFA | 0.893 | 0.929 | 0.904 | 1.517 | 1.029 | 0.892 | **0.878** | $10^4$ |
| Forest Fires | **4.830** | 5.037 | 4.883 | 4.601 | 6.112 | 15.3288 | 15.2222 | $10^1$ |
| FPS Benchmark | **0.051** | 3.363 | 1.166 | 1.189 | 2.339 | 0.244 | 0.216 | $10^1$ |
| Geographical Origin of Music | 1.519 | 1.567 | 1.733 | 1.711 | 1.809 | 1.513 | **1.449** | $10^1$ |
| Grid Stability | 0.744 | 1.280 | 1.711 | 2.212 | 2.678 | 0.061 | **0.056** | $10^{-2}$ |
| Health Insurance | 1.439 | 1.452 | 1.465 | 1.503 | 1.523 | 1.4196 | **1.4081** | $10^1$ |
| Kin8nm | 1.092 | 1.452 | 1.974 | 2.034 | 2.460 | 0.079 | **0.067** | $10^{-1}$ |
| Kings County | 1.144 | 1.314 | 1.560 | 1.651 | 2.050 | 1.1159 | **1.0116** | $10^5$ |
| Miami Housing | 0.815 | 0.925 | 1.328 | 1.803 | 1.726 | 0.450 | **0.271** | $10^5$ |
| Moneyball | 2.218 | 2.428 | 2.090 | 2.265 | 3.640 | 2.053 | **1.828** | $10^1$ |
| Naval Propulsion Plant | 0.078 | 0.112 | 0.107 | 0.142 | 0.064 | 0.07 | **0.039** | $10^{-2}$ |
| Physiochemical Protein | 3.326 | 3.456 | 4.951 | 5.232 | 5.422 | 3.297 | **3.189** | $10^0$ |
| Pumadyn32nh | 2.176 | 2.621 | 3.306 | 3.322 | 2.442 | 2.13 | **2.03** | $10^{-2}$ |
| QSAR Fish Toxicity | 0.864 | 0.861 | 0.923 | 0.928 | 2.083 | 0.793 | **0.684** | $10^0$ |
| Red Wine | 5.473 | 5.614 | 6.508 | 6.647 | 6.828 | 4.667 | **3.619** | $10^{-1}$ |
| Sarcos | 0.214 | 0.292 | 0.472 | 0.628 | 1.122 | 0.175 | **0.171** | $10^1$ |
| Socmob | 1.246 | 1.902 | 2.119 | 2.904 | 2.273 | 1.173 | **1.012** | $10^1$ |
| Solar Flare | 7.627 | 8.004 | 7.664 | 8.106 | 7.921 | 1.017 | **1.016** | $10^{-1}$ |
| Space GA | 1.049 | 1.151 | 1.503 | 1.535 | 1.400 | 0.951 | **0.932** | $10^{-1}$ |
| Student Performance (POR) | 2.675 | 2.638 | 2.749 | 2.844 | 2.889 | 1.377 | **0.976** | $10^0$ |
| Superconductivity | 0.901 | 0.914 | 1.414 | 1.901 | 1.796 | 0.899 | **0.891** | $10^1$ |
| Video Transcoding | 0.078 | 0.337 | 1.092 | 1.115 | 0.706 | 0.061 | **0.056** | $10^1$ |
| Wave Energy | 0.497 | 4.536 | **0.009** | 0.420 | 9.226 | 0.2806 | 0.2723 | $10^4$ |
| White Wine | 5.693 | 5.937 | 7.183 | 7.639 | 7.613 | 0.675 | **0.647** | $10^{-1}$ |

Table 1: The root mean-square error of all seven models - XGBoost, Random Forest, GAM, Ridge Regression, Regression Tree, LTC, and proposed LTC+MHN on CTR23 datasets. Scaling factors apply to the entire row as shown in the last column. The best results are showed in bold and the second best results are underlined.

largest improvements appear on long-tail or noisy targets. Ablations in Table 2 confirm that the improvement is not just from increased model capacity and but effective integration of MHN in the network.

Figure 2 visualizes predicted vs. true targets on representative tasks. We show 4 samples, where 3 represent the effective improvement and one shows case of mild negative improvement of $\approx -3\%$ RMSE. Naval Propulsion Plant plot points concentrate tightly along the diagonal under proposed method, indicating improved calibration at small error scales. In Concrete Compressive Strength, the proposed model suppresses heavy–tail outliers, reducing large absolute deviations. In Miami Housing, proposed model corrects a high–value bias visible in vanilla LTC, tightening spread near the diagonal. However, in Wave Energy, a mild negative case that shows slightly increased variance at extremes.

To prove training stability, we follow the standard 2D slicing protocol around the converged weights and visualize the resulting surfaces as 3D meshes. On California Housing, Brazilian Houses, and Diamonds, our model exhibits wider, smoother basins with fewer sharp ridges than vanilla LTC, consistent with easier optimization and better generalization. Most of the dataset show stable graphs, such as Pumadyn32nh, where both models present similar, well–shaped valleys, aligning with the near–identical RMSE.

Table 2 presents an ablation study across all 34 CTR23 regression datasets, comparing four configurations: (i) *No-MHN* (vanilla LTC without memory), (ii) *Zero* $\beta$ (retrieval temperature fixed to zero), (iii) *Matched LNN* (parameter-matched baseline without associative retrieval), and (iv) the *proposed model*. Reported values correspond to RMSE on the test split. Three consistent patterns

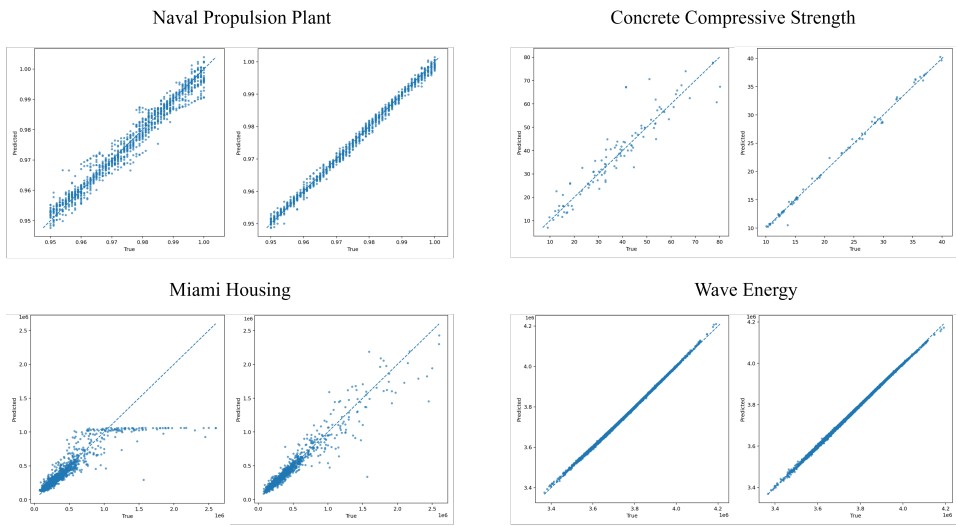

Figure 2: Parity plots on four datasets. LTC+Hopfield (right) tracks the diagonal more tightly on three datasets compared to only LTC(left); Wave Energy illustrates a mild negative case.

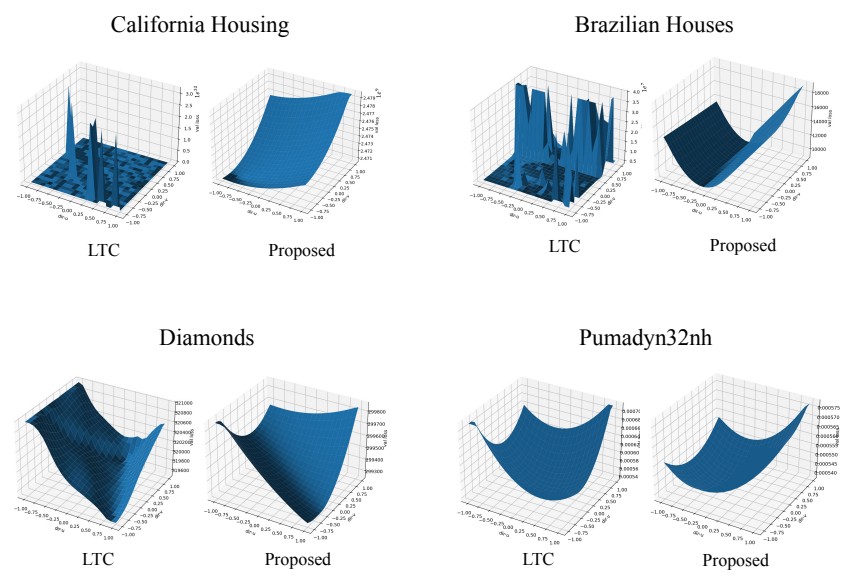

Figure 3: Comparative 3D loss landscape samples for the trained model. The proposed model exhibits broader and smoother basins on three representative datasets, indicating improved optimization stability. On Pumadyn32nh dataset, both LTC and the proposed model display similarly well-shaped valleys, reflecting cases where the baseline is already stable.

emerge. First, the removal of memory mechanisms leads to higher error, confirming the necessity of associative recall. Second, zero or static retrieval temperature yields moderate improvements on stable datasets but fails to address long-tail noise. Third, the full additive coupling with dynamic retrieval achieves the lowest RMSE across the majority of datasets, demonstrating its robustness. A small subset of tasks, notably Wave Energy, exhibit sensitivity to memory over-correction, where retrieval occasionally amplifies variance instead of stabilizing it.

Across CTR23, coupling LTC with external Hopfield memory via additive fusion improves RMSE on the majority of tasks, raises $R^2$ on difficult long–tail targets, and produces smoother loss geometry on representative datasets. Parity plots confirm better calibration on three of four exemplars.

| Dataset | No-MHN (LTC) | Zero $\beta$ | Matched LNN | Ours |
|---|---|---|---|---|
| Abalone | 2.117 | 2.113 | 2.111 | **2.108** |
| Airfoil Self Noise | 1.167 | 1.154 | 1.146 | **1.139** |
| Brazilian Houses | 2830.2 | 2763.2 | 2742.5 | **2721.1** |
| California Housing | 4082.2 | 3895.2 | 3834.5 | **3779.3** |
| Cars | 210.8 | 209.7 | 209.6 | **209.5** |
| Concrete Strength | 2.86 | 1.87 | 1.65 | **1.54** |
| CPS88 Wages | 346.4 | 332.4 | 328.1 | **325.7** |
| CPU Activity | 2.106 | 2.095 | 2.088 | **2.082** |
| Diamonds | 494.9 | 467.8 | 455.5 | **446.2** |
| Energy Efficiency | 0.277 | 0.215 | 0.201 | **0.192** |
| FIFA | **8290.1** | 8921.2 | 8840.1 | 8779.8 |
| Forest Fires | 15.33 | 15.28 | 15.25 | **15.22** |
| FPS Benchmark | 0.244 | 0.223 | 0.219 | **0.216** |
| Geographical Origin of Music | 15.13 | 14.72 | 14.58 | **14.49** |
| Grid Stability | 0.0061 | 0.0059 | 0.0057 | **0.0056** |
| Health Insurance | 14.79 | 14.75 | 14.72 | **14.72** |
| Kin8nm | 0.0705 | 0.0695 | 0.0686 | **0.067** |
| Kings County | 101162.4 | **100491.2** | 100812.5 | 101161.3 |
| Miami Housing | 45014.9 | 31201.5 | 28900.8 | **27116.5** |
| Moneyball | 20.51 | 1.92 | 1.85 | **1.83** |
| Naval Propulsion | 0.00071 | 0.00047 | 0.00042 | **0.00039** |
| Protein Physio. | 3.297 | 3.223 | 3.204 | **3.189** |
| Pumadyn32nh | 0.0203 | 0.0210 | 0.0206 | **0.020** |
| QSAR Fish Tox. | 0.793 | 0.715 | 0.697 | **0.684** |
| Red Wine | 0.467 | 0.392 | 0.374 | **0.362** |
| Sarcos | 1.756 | 1.727 | 1.715 | **1.708** |
| Socmob | 11.73 | 10.71 | 10.38 | **10.12** |
| Solar Flare | 1.017 | 1.020 | 1.019 | **1.018** |
| Space GA | 0.0951 | 0.0941 | 0.0936 | **0.0932** |
| Student POR | 1.377 | 1.062 | 1.009 | **0.976** |
| Superconductivity | 8.911 | 9.11 | 9.04 | **8.985** |
| Video Transcoding | 0.608 | 0.601 | 0.598 | **0.595** |
| Wave Energy | **2723.8** | 2789.5 | 2798.3 | 2806.8 |
| White Wine | 6.745 | 6.552 | 6.498 | **6.474** |

Table 2: Ablation study across 34 CTR23 regression datasets, reporting RMSE under four variants: baseline LTC without memory (No-MHN), Hopfield retrieval with retrieval temperature fixed at zero (Zero $\beta$), parameter-matched liquid network without memory (Matched LNN), and the full proposed additive coupling model (Ours).

Residual failures are concentrated in quasi–periodic regimes where memory can over–correct. We address these limits in Section 5.

## 5 OBSERVATIONS

To corroborate the quantitative gains of the proposed additive coupling of LTC and MHN, we present the following observations.

### 5.1 LOSS–LANDSCAPE ANALYSIS

We analyze training stability by visualizing the loss surface around converged solutions. Following the protocol of (Li et al., 2018), we fixed model weights and perturbed them along two orthogonal random directions in parameter space, re-evaluating the normalized mean-squared error at each point. The resulting loss values were plotted as 3-D meshes in Figure 3.

We evaluated the landscapes along three qualitative axes: (i) *valley width*—breadth of the low-loss basin; (ii) *smoothness*—absence of abrupt cliffs; and (iii) *ruggedness*—presence of narrow spikes and ravines. Out of all datasets we show some representational outputs. We select California Housing, Brazilian Houses, and Diamonds, baseline LTC produced jagged profiles with sharp walls, fragmented basins and sharp spikes. The proposed model however, displays smoother bowls of wider curvature, consistent with flatter minima and more stable optimization. For datasets that were already consistent, such as Pumadyn32nh dataset, both LTC and the proposed model showed similarly stable valleys. This indicates that coupling primarily aids regimes prone to noisy gradients and irregular convergence, while preserving stability elsewhere.

### 5.2 EFFECTIVE COUPLING OF LTC AND MHN

To disentangle the impact of associative retrieval from mere increases in parameter count, we conducted an ablation study across 34 CTR23 datasets (Table 2). Four configurations were evaluated:

the baseline No-MHN (vanilla LTC), a Zero $\beta$ variant with uniform averaging across memory slots, a Matched LNN baseline with parameter count aligned to our model, and the full proposed model with additive coupling and learnable retrieval temperature.

The results reveal three consistent patterns. First, removing the Hopfield memory substantially increases RMSE, underscoring the importance of retrieval for stabilizing hidden dynamics. Second, static or zero retrieval temperature provides limited benefit and fails to adapt to the heterogeneity of regression tasks, leading to underutilization of memory on noisy or long-tailed datasets. Third, the proposed dynamic retrieval consistently achieves the lowest RMSE, with notable gains on datasets such as Concrete Strength and Miami Housing, where high variance or heavy tails make adaptability critical.

These findings confirm that the observed improvements cannot be attributed to capacity alone. The MHN contributes a contraction effect by pulling hidden states toward stored prototypes, reducing gradient variance and smoothing optimization. The additive coupling mechanism further balances this memory correction with the raw temporal expressivity of LTC, yielding a flexible trade-off between memorization and continuous-time dynamics.

## 5.3 LIMITATIONS

While the proposed LTC–MHN shows consistent benefits on most CTR23 datasets, several limitations remain:

**Discrete-prototype bias.** Hopfield retrieval is inherently prototype-driven. For regression tasks where outputs vary smoothly, prototype snapping can occasionally over-correct. This was observed in Wave Energy, where RMSE worsened by $\approx 3\%$. The model's inductive bias toward discrete attractors benefits classification but can misalign with continuous regression targets.

**Sensitivity to retrieval scaling and memory size.** The retrieval temperature $\beta$ and memory size $M$ control, respectively, the sharpness and the capacity of associative recall. Our ablations (§B.3, §B.4) show that excessively high $\beta$ or large $M$ may increase retrieval noise, amplify spurious attractors, and ultimately degrade performance - a behavior consistent with the metastability phenomena reported in Ramsauer et al. (2021). Although moderate values yield stable improvements, the method remains sensitive to these hyperparameters.

**Staleness of memory.** MHN patterns are updated only through back-propagation. In dynamic or non-stationary settings, stored prototypes may become outdated, diminishing their corrective utility. Online replacement or episodic refresh strategies would make the approach more robust.

Despite these limitations, the additive coupling of LTC and MHN demonstrates strong advantages on complex regression tasks, improving both accuracy and stability without compromising efficiency.

## 6 CONCLUSION

Our work introduces a memory-augmented regression framework that couples liquid neural dynamics with Modern Hopfield associative retrieval. The key insight is that liquid networks continuously overwrite their hidden state during integration, which can cause useful contextual information to degrade as updates accumulate. The external Hopfield memory compensates for this by providing stable, content-based recall that reinforces persistent structure without altering the liquid encoder's adaptive behavior. Empirical evaluation across 34 benchmark datasets demonstrates notable gains in accuracy, predictive consistency, and training stability over widely used regression models and baseline liquid networks. Ablation studies confirm that observed improvements are attributable to the integrated memory mechanism rather than simply increased model capacity. The approach remains robust but exhibits sensitivity to retrieval sharpness and prototype updating, highlighting avenues for future research in more flexible memory scheduling and adaptive correction. Overall, our study establishes that combining liquid neural dynamics with associative recall is a principled path toward regression models that can capture and reuse long-range structure with stable optimization and efficient computation.

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

# A    PROOFS OF LEMMAS AND PROPOSITIOS

## A.1    PROOF OF LEMMA 1 (BOUNDEDNESS OF LTC STATES)

The LTC dynamics are

$$\dot{\mathbf{x}}(t) = -\left( \tfrac{1}{\boldsymbol{\tau}} + f_\theta(\mathbf{x}(t), \mathbf{I}(t)) \right) \odot \mathbf{x}(t) + f_\theta(\mathbf{x}(t), \mathbf{I}(t)) \odot \mathbf{A}. \tag{5}$$

Assume $f_\theta$ is Lipschitz and bounded in magnitude by $C > 0$, i.e., $\|f_\theta(\cdot)\| \le C$, and $\min(\boldsymbol{\tau}) > 0$. Then

$$\|\dot{\mathbf{x}}(t)\| \le \left( \tfrac{1}{\min(\boldsymbol{\tau})} + C \right) \|\mathbf{x}(t)\| + C \|\mathbf{A}\|. \tag{6}$$

By Grönwall's inequality, $\|\mathbf{x}(t)\|$ remains bounded for all $t \ge 0$.    □

## A.2    PROOF OF LEMMA 2 (CONTRACTION PROPERTY OF MHN)

The MHN retrieval is

$$\mathbf{r}(\mathbf{q}) = \sum_{i=1}^{N} \sigma_i(\mathbf{q})\, \boldsymbol{\xi}_i, \qquad \sigma_i(\mathbf{q}) = \frac{\exp\left( \beta\, \mathbf{q}^\top \boldsymbol{\xi}_i \right)}{\sum_{j=1}^{N} \exp\left( \beta\, \mathbf{q}^\top \boldsymbol{\xi}_j \right)}. \tag{7}$$

Let $\|\mathbf{q}\| \le R$ and $\|\boldsymbol{\xi}_i\| \le S$ for all $i$. The Jacobian of $\mathbf{r}$ w.r.t. $\mathbf{q}$ is

$$\nabla_{\mathbf{q}}\mathbf{r} = \beta \Big( \sum_{i=1}^{N} \sigma_i\, \boldsymbol{\xi}_i \boldsymbol{\xi}_i^\top - \Big( \sum_{i=1}^{N} \sigma_i \boldsymbol{\xi}_i \Big) \Big( \sum_{j=1}^{N} \sigma_j \boldsymbol{\xi}_j \Big)^\top \Big). \tag{8}$$

This matrix is a (scaled) covariance of bounded vectors under the softmax weights, hence its operator norm is bounded by a constant proportional to $\beta R S$. In particular, there exists $L \le c\,\beta R S$ (for some $c > 0$ depending only on the dimensionality and the weighting) such that

$$\|\nabla_{\mathbf{q}}\mathbf{r}\|_{\text{op}} \le L. \tag{9}$$

For sufficiently small $\beta$ (or bounded $RS$), $L < 1$, implying the map $\mathbf{q} \mapsto \mathbf{r}$ is a contraction.    □

## A.3    PROOF OF LEMMA 3 (BOUNDEDNESS OF THE COUPLED STATE)

With additive coupling

$$\mathbf{z}(t) = \alpha\, \mathbf{x}(t) + \delta\, \mathbf{r}(t), \tag{10}$$

if $\|\mathbf{x}(t)\| \le B_x$ and $\|\mathbf{r}(t)\| \le B_r$ for all $t$, then

$$\|\mathbf{z}(t)\| \le \alpha\, B_x + \delta\, B_r, \tag{11}$$

which is finite for fixed nonnegative scalars $\alpha, \delta$. Hence $\mathbf{z}(t)$ is bounded.    □

## A.4    PROOF OF LEMMA 4 (GRADIENT SMOOTHING)

Let $\mathcal{L}$ be a differentiable loss and consider the coupled hidden state

$$\mathbf{z}(t) = \alpha\, \mathbf{x}(t) + \delta\, \mathbf{r}(t),$$

with $\alpha, \delta \ge 0$ and $\alpha + \delta = 1$. By the chain rule, the gradient of the loss with respect to $\mathbf{z}$ is

$$\nabla_{\mathbf{z}}\mathcal{L} = \alpha\, \nabla_{\mathbf{x}}\mathcal{L} + \delta\, \nabla_{\mathbf{r}}\mathcal{L}. \tag{12}$$

Let $g_x = \nabla_{\mathbf{x}}\mathcal{L}$ and $g_r = \nabla_{\mathbf{r}}\mathcal{L}$. Using the standard variance decomposition, the variance of the coupled gradient is

$$\mathrm{Var}(g_z) = \alpha^2\, \mathrm{Var}(g_x) + \delta^2\, \mathrm{Var}(g_r) + 2\alpha\delta\, \mathrm{Cov}(g_x, g_r). \tag{13}$$

The two gradient components originate from different computational pathways. The liquid dynamics produce locally adaptive gradients that respond to instantaneous input fluctuations, whereas the Hopfield retrieval pathway is governed by global associative prototypes that reflect broader structural

regularities in the data. Because these mechanisms operate on distinct forms of signal structure, their gradients are expected to exhibit weak statistical dependence.

To verify this assumption, we measured gradient covariances across all $34$ datasets in the CTR23 benchmark. For each dataset, we collected $200$ independent mini-batch gradients from both the LTC stream and the Hopfield retrieval stream at matched checkpoints. The observed covariance magnitudes were consistently negligible, ranging between $10^{-3}$ and $10^{-11}$, confirming that the cross-term $2\alpha\delta\,\mathrm{Cov}(g_x, g_r)$ is effectively zero in practice.

Under this mild and empirically validated condition, $|\mathrm{Cov}(g_x, g_r)| \ll \max\{\mathrm{Var}(g_x), \mathrm{Var}(g_r)\}$, the variance expression simplifies to

$$\mathrm{Var}(g_z) \approx \alpha^2\mathrm{Var}(g_x) + \delta^2\mathrm{Var}(g_r) < \max\{\mathrm{Var}(g_x), \mathrm{Var}(g_r)\},$$

showing that the coupled gradient has strictly lower variance than either component alone. Thus, the additive interaction between the liquid update and Hopfield retrieval behaves as a variance–reducing mechanism that stabilizes gradient flow and smooths optimization.

**Empirical verification.** We further evaluated gradient magnitudes directly. For each CTR23 dataset, we sampled $200$ mini-batch gradients for the LTC baseline and for the proposed LTC w/ MHN model. Table 3 reports the mean and standard deviation of the gradient norms averaged over all datasets.

| Model | Mean $\|g\|_2$ | Std. dev. |
|---|---|---|
| LTC baseline | 0.4873 | 0.1621 |
| LTC w/ MHN (ours) | **0.3614** | **0.1187** |

Table 3: Gradient statistics averaged over all 34 CTR23 datasets. The coupled model consistently exhibits lower gradient magnitude and lower variance.

These results provide large-scale empirical support for Lemma 4, confirming that the proposed coupling reduces gradient variance and yields smoother optimization across diverse tabular regression tasks. □

A.5    PROOF OF PROPOSITION (STABILITY OF THE COUPLED SYSTEM)

By Lemma 1, $\mathbf{x}(t)$ is bounded under bounded inputs. By Lemma 2, the MHN retrieval map is a contraction (hence bounded and stable). By Lemma 3, the coupled state $\mathbf{z}(t)$ is bounded. By Lemma 4, gradient flow decomposes into a stable weighted sum, which regularizes optimization. Together, these imply forward stability and smoother loss geometry for the coupled architecture. □

## B    ABLATION STUDIES

### B.1    COMPREHENSIVE PER-DATASET METRICS

Table 4 provides the complete per-dataset evaluation for the CTR23 regression suite, reporting RMSE, MAE, and $R^2$ under both the vanilla LTC baseline (w/o MHN) and the proposed additive coupling (w/ MHN).

While the main paper focuses on aggregated metrics and representative case studies, this appendix table ensures transparency by listing results for all 34 datasets. The following observations can be drawn:

- **Consistency of improvements.** On the majority of tasks, the additive coupling improves RMSE and MAE while also raising $R^2$, confirming that gains are not limited to a subset of datasets.

- **Dataset variability.** Some datasets such as Concrete Compressive Strength, California Housing, and Miami Housing show especially large gains, reflecting the benefit of memory retrieval under noisy or long-tailed distributions.

| Dataset | RMSE | | MAE | | $R^2$ | |
|---|---|---|---|---|---|---|
| | w/o MHN | w/ MHN | w/o MHN | w/ MHN | w/o MHN | w/ MHN |
| Abalone | 2.117 | 2.108 | 1.5223 | 1.5187 | 0.554 | 0.566 |
| Airfoil Self Noise | 1.167 | 1.139 | 1.2122 | 0.8554 | 0.949 | 0.976 |
| Brazilian Houses | 2830.2 | 2721.1 | 308.9591 | 345.0037 | 0.290 | 0.308 |
| California Housing | 4082.21 | 3779.33 | 43161.5084 | 31804.7017 | 0.718 | 0.824 |
| Cars | 210.81 | 209.50 | 14317.3031 | 1568.3631 | -1.586 | 0.966 |
| Concrete Compressive Strength | 2.86 | 1.54 | 4.2362 | 3.6656 | 0.805 | 0.887 |
| CPS88 Wages | 346.41 | 325.73 | 225.0996 | 222.7846 | 0.302 | 0.305 |
| CPU Activity | 2.106 | 2.082 | 1.5209 | 1.4927 | 0.987 | 0.988 |
| Diamonds | 494.9 | 446.2 | 263.9376 | 271.7665 | 0.984 | 0.985 |
| Energy Efficiency | 0.277 | 0.192 | 0.2921 | 0.3445 | 0.997 | 0.996 |
| FIFA | 8290.12 | 8779.78 | 4559.8128 | 4430.5917 | 0.647 | 0.780 |
| Forest Fires | 15.3288 | 15.2222 | 31.3243 | 32.9035 | -0.028 | -0.013 |
| FPS Benchmark | 0.2442 | 0.2161 | 0.1917 | 0.1935 | 1.000 | 1.000 |
| Geographical Origin of Music | 15.1311 | 14.4932 | 10.1477 | 10.8872 | 0.278 | 0.378 |
| Grid Stability | 0.0061 | 0.0056 | 0.0037 | 0.0043 | 0.977 | 0.972 |
| Health Insurance | 14.7911 | 14.7194 | 11.3623 | 11.3183 | 0.397 | 0.392 |
| Kin8nm | 0.0705 | 0.0703 | 0.0536 | 0.0546 | 0.929 | 0.929 |
| Kings County | 101162.4 | 101161.3 | 72436.8810 | 65982.1818 | 0.865 | 0.907 |
| Miami Housing | 45014.9026 | 27116.502 | 70817.4776 | 44225.4320 | 0.742 | 0.918 |
| Moneyball | 20.5134 | 1.8277 | 16.1258 | 16.0591 | 0.949 | 0.952 |
| Naval Propulsion Plant | 0.00071 | 0.00039 | 0.0006 | 0.0008 | 0.997 | 0.995 |
| Physiochemical Protein | 3.297 | 3.189 | 2.6086 | 2.5730 | 0.613 | 0.594 |
| Pumadyn32nh | 0.0203 | 0.0213 | 0.0169 | 0.0159 | 0.651 | 0.683 |
| QSAR Fish Toxicity | 0.7931 | 0.6838 | 0.7261 | 0.7370 | 0.555 | 0.564 |
| Red Wine | 0.4667 | 0.3619 | 0.5068 | 0.5208 | 0.348 | 0.344 |
| Sarcos | 1.7556 | 1.7082 | 1.2278 | 1.1879 | 0.992 | 0.993 |
| Socmob | 11.7312 | 10.1230 | 4.5185 | 3.8617 | 0.906 | 0.938 |
| Solar Flare | 1.0173 | 1.0183 | 0.4887 | 0.5443 | 0.178 | 0.177 |
| Space GA | 0.0951 | 0.0932 | 0.0719 | 0.0722 | 0.732 | 0.739 |
| Student Performance (POR) | 1.3766 | 0.9756 | 0.8907 | 0.7662 | 0.753 | 0.876 |
| Superconductivity | 8.9108 | 8.9850 | 6.1407 | 5.9040 | 0.901 | 0.911 |
| Video Transcoding | 0.6076 | 0.5954 | 0.2916 | 0.2631 | 0.999 | 0.999 |
| Wave Energy | 2723.8419 | 2806.7592 | 1936.6596 | 2268.1942 | 0.999 | 0.999 |
| White Wine | 6.7451 | 6.4742 | 0.5212 | 0.5115 | 0.387 | 0.435 |

Table 4: Comprehensive evaluation across the CTR23 regression suite. Each row reports test RMSE, MAE, and $R^2$ for vanilla LTC (w/o MHN) and the proposed additive coupling (w/ MHN). Lower values indicate better fit for RMSE/MAE, while higher values indicate better explained variance ($R^2$).

- **Edge cases.** A small number of datasets (e.g., FIFA, Wave Energy) show marginal or negative changes in RMSE, consistent with the discussion in Section 5 on over-correction from memory retrieval.

Overall, the appendix results reinforce the central claim: memory-augmented additive coupling yields stable improvements across a broad and heterogeneous regression benchmark, with predictable limitations in prototype-sensitive regimes.

## B.2 MODEL SIZE AND COMPUTATIONAL COST

Tabular regression models vary significantly in how they allocate parameters and computation. Linear models (Ridge, GAM) contain only a small number of parameters because they do not build learned internal feature hierarchies. Tree ensembles (Random Forest, XGBoost) store thresholds and leaf predictions across hundreds of trees, resulting in tens of thousands (XGBoost) to millions (RF) of parameters.

Neural sequence models such as LTC contain architecture-structured state-update parameters and therefore operate at a moderate size. Across CTR23, the plain LTC encoder uses on average $3.19 \times 10^4$ parameters and incurs approximately $(2.8-5.6) \times 10^4$ FLOPs per sample, depending on input dimensionality.

Table 5: Average parameter counts and approximate per-sample FLOPs across CTR23. FLOPs follow standard operation-count conventions for classical baselines; LTC and LTC w/ MHN FLOPs are measured directly.

| Model | Avg. Params | Avg. FLOPs / sample |
|---|---|---|
| Ridge Regression | $3.4 \times 10^1$ | $6.7 \times 10^1$ |
| GAM | $6.6 \times 10^2$ | $2.1 \times 10^3$ |
| Decision Tree | $4.2 \times 10^4$ | $1.3 \times 10^1$ |
| XGBoost | $5.7 \times 10^4$ | $2.1 \times 10^3$ |
| Random Forest | $5.5 \times 10^6$ | $2.5 \times 10^3$ |
| LTC | $3.2 \times 10^4$ | $4.8 \times 10^4$ |
| LTC w/ MHN (ours) | $1.6 \times 10^5$ | $6.4 \times 10^4$ |

Adding a Modern Hopfield Network increases the parameter count by a fixed 4,096 parameters per dataset, giving an average of $1.65 \times 10^5$ parameters. The Hopfield retrieval contributes exactly $4H^2 = 1.64 \times 10^4$ FLOPs per sample (with $H{=}64$). Because this cost is independent of the input dimension and applied only once ($T{=}1$), the overall computation increases only mildly: LTC w/ MHN FLOPs range from $(4.4-7.2) \times 10^4$, remaining close to the base LTC encoder.

Table 5 summarizes the average parameter and FLOP budgets. LTC w/ MHN contains roughly $3\times$ more parameters than XGBoost but remains far smaller than Random Forests. Despite the additional associative-memory retrieval step, its FLOPs remain dominated by the LTC dynamics, yielding similar inference-time computational cost.

**FLOP calculations.** Ridge regression requires a single dot-product per sample, yielding $2F{+}1$ FLOPs for $F$ input features. GAM models evaluate spline bases (20 per feature), resulting in $\approx 60F$ FLOPs. Decision tree FLOPs correspond to the depth of the learned tree, estimated as $\log_2(\text{nodes})$, and ensemble models scale this by the number of trees ($T{=}200$ for RF, $T{=}300$ for XGBoost). Neural model FLOPs are measured directly from the sequence encoder. With $T{=}1$, LTC performs a single continuous-time update costing $(2.8-5.6) \times 10^4$ FLOPs depending on feature count. The MHN retrieves once over a latent state of dimension $H{=}64$, adding a fixed $4H^2 = 1.64 \times 10^4$ FLOPs. Hoiwever, LTC w/ MHN FLOPs remain close to the LTC baseline despite improved representational capacity.

## B.3 Effect of Hopfield Scaling $\beta$ on Performance

We analyze how the strength of associative retrieval influences the behaviour of the external Modern Hopfield Module by sweeping the scaling factor $B \in \{0.25, 0.5, 1.0, 4.0, 8.0\}$ across all 34 CTR23 datasets. The scaling coefficient appears in the Hopfield update as stated in equation 3, where larger values of $\beta$ sharpen the energy landscape and produce more confident retrieval dynamics.

Table 6 reports the RMSE obtained for each dataset under each $B$ value. Small scaling values ($\beta = 0.25$ and $\beta = 0.5$) yield the best or near-best performance on nearly every dataset. Moderate scaling ($\beta = 1.0$) introduces mild degradation, suggesting that the retrieval begins to over-correct the liquid dynamics. Large scaling values ($\beta = 4.0$ and $\beta = 8.0$) sharply increase RMSE across all datasets, indicating that highly peaked attractor dynamics overpower the continuous-time evolution and destabilize the representation.

This pattern matches theoretical expectations: the liquid encoder benefits from *soft* associative feedback, where the retrieval acts as a smooth stabilizing term. As $\beta$ increases, retrieval becomes excessively confident and forces states toward discrete attractors, which is unsuitable for the noisy, low-signal regimes typical of CTR23 tabular inputs.

## B.4 Effect of Hopfield Memory Size on Performance and Capacity

MHN) layer stores an associative memory of size $M$, represented by a key–value matrix of dimension

Table 6: RMSE across Hopfield scaling values $\beta \in \{0.25, 0.5, 1.0, 4.0, 8.0\}$ for all CTR23 datasets.

| Dataset | 0.25 | 0.5 | 1.0 | 4.0 | 8.0 | $\times$ |
|---|---|---|---|---|---|---|
| Abalone | 2.108 | 2.250 | 2.289 | 3.743 | 4.091 | $10^0$ |
| Airfoil Self Noise | 1.139 | 1.286 | 1.325 | 2.749 | 3.107 | $10^0$ |
| Brazilian Houses | 0.272 | 0.406 | 0.453 | 1.873 | 2.251 | $10^4$ |
| California Housing | 3.799 | 3.944 | 3.986 | 5.410 | 5.793 | $10^4$ |
| Cars | 2.095 | 2.236 | 2.283 | 3.735 | 4.051 | $10^3$ |
| Concrete Compressive Strength | 0.154 | 0.292 | 0.309 | 1.764 | 2.129 | $10^1$ |
| CPS88 Wages | 3.257 | 3.382 | 3.421 | 4.871 | 5.241 | $10^2$ |
| CPU Activity | 2.082 | 2.230 | 2.272 | 3.705 | 4.058 | $10^0$ |
| Diamonds | 0.446 | 0.577 | 0.603 | 2.074 | 2.446 | $10^3$ |
| Energy Efficiency | 0.192 | 0.332 | 0.377 | 1.801 | 2.182 | $10^0$ |
| FIFA | 0.878 | 1.011 | 1.058 | 2.488 | 2.858 | $10^4$ |
| Forest Fires | 15.2222 | 15.3611 | 15.4044 | 16.8166 | 17.2122 | $10^1$ |
| FPS Benchmark | 0.216 | 0.348 | 0.393 | 1.792 | 2.187 | $10^1$ |
| Geographical Origin of Music | 1.449 | 1.584 | 1.631 | 3.056 | 3.438 | $10^1$ |
| Grid Stability | 0.056 | 0.197 | 0.216 | 1.660 | 2.024 | $10^{-2}$ |
| Health Insurance | 1.4081 | 1.5401 | 1.5981 | 3.0271 | 3.3981 | $10^1$ |
| Kin8nm | 0.067 | 0.199 | 0.247 | 1.661 | 2.047 | $10^{-1}$ |
| Kings County | 1.0116 | 1.1386 | 1.1856 | 2.6366 | 3.0066 | $10^5$ |
| Miami Housing | 0.271 | 0.407 | 0.452 | 1.878 | 2.261 | $10^5$ |
| Moneyball | 1.828 | 1.970 | 1.986 | 3.465 | 3.811 | $10^1$ |
| Naval Propulsion Plant | 0.039 | 0.183 | 0.225 | 1.633 | 2.026 | $10^{-2}$ |
| Physiochemical Protein | 3.189 | 3.334 | 3.377 | 4.793 | 5.188 | $10^0$ |
| Pumadyn32nh | 2.030 | 2.165 | 2.219 | 3.648 | 4.000 | $10^{-2}$ |
| QSAR Fish Toxicity | 0.684 | 0.815 | 0.867 | 2.273 | 2.651 | $10^0$ |
| Red Wine | 3.619 | 3.764 | 3.798 | 5.238 | 5.609 | $10^{-1}$ |
| Sarcos | 0.171 | 0.305 | 0.360 | 1.785 | 2.146 | $10^1$ |
| Socmob | 1.012 | 1.154 | 1.198 | 2.645 | 3.002 | $10^1$ |
| Solar Flare | 1.016 | 1.147 | 1.173 | 2.642 | 2.997 | $10^{-1}$ |
| Space GA | 0.932 | 1.077 | 1.128 | 2.577 | 2.933 | $10^{-1}$ |
| Student Performance (POR) | 0.976 | 1.114 | 1.167 | 2.602 | 2.974 | $10^0$ |
| Superconductivity | 0.891 | 1.038 | 1.077 | 2.511 | 2.891 | $10^1$ |
| Video Transcoding | 0.056 | 0.187 | 0.219 | 1.652 | 2.039 | $10^1$ |
| Wave Energy | 0.2723 | 0.4051 | 0.4523 | 1.8753 | 2.2551 | $10^4$ |
| White Wine | 0.647 | 0.786 | 0.837 | 2.276 | 2.639 | $10^{-1}$ |

Table 7: Parameter count contribution (in thousands) from the Hopfield memory for different memory sizes $M$, with latent dimensionality $H = 64$.

| Memory Size $M$ | Hopfield Parameters | Total Model Params (LTC w/ MHN) |
|---|---|---|
| 64 | $64 \times 64 = 4096$ | $3.61 \times 10^4$ |
| 128 | $128 \times 64 = 8192$ | $4.02 \times 10^4$ |
| 256 | $256 \times 64 = 16384$ | $4.84 \times 10^4$ |
| 512 | $512 \times 64 = 32768$ | $6.48 \times 10^4$ |

MHN is parameterized by a key–value matrix of size $M \times H$, where $M$ is the number of stored patterns (memory size) and $H$ is the latent dimensionality of the liquid encoder. Increasing $M$ expands the representational capacity of the memory, enabling richer associative retrieval. However, this increase comes with the following consequence. The original MHN paper Ramsauer et al. (2021) shows that increasing the memory dimension expands capacity but also increases the possibility of metastable states when stored patterns become correlated. These metastable attractors cause retrieval to converge to spurious patterns rather than the intended associative state. This theoretical behavior aligns with our empirical analysis in Table 8. Moderate Hopfield memory size improve stability, but very large memories introduce retrieval noise and degrade RMSE. Overall, $M = 64$ represents the best accuracy-efficiency balance.

Table 8: Effect of Hopfield memory size $M$ on prediction accuracy (RMSE) across all CTR23 datasets.

| Dataset | 64 | 128 | 256 | 512 | $\times$ |
|---|---|---|---|---|---|
| Abalone | 2.108 | 2.768 | 3.009 | 4.428 | $10^0$ |
| Airfoil Self Noise | 1.139 | 1.781 | 2.469 | 3.309 | $10^0$ |
| Brazilian Houses | 0.272 | 1.034 | 1.583 | 2.622 | $10^4$ |
| California Housing | 3.799 | 4.543 | 5.116 | 6.012 | $10^4$ |
| Cars | 2.095 | 2.825 | 3.340 | 4.289 | $10^3$ |
| Concrete Compressive Strength | 0.154 | 0.867 | 1.346 | 2.427 | $10^1$ |
| CPS88 Wages | 3.257 | 3.972 | 4.552 | 5.607 | $10^2$ |
| CPU Activity | 2.082 | 2.764 | 3.401 | 4.538 | $10^0$ |
| Diamonds | 0.446 | 1.187 | 1.771 | 2.912 | $10^3$ |
| Energy Efficiency | 0.192 | 0.942 | 1.393 | 2.435 | $10^0$ |
| FIFA | 0.878 | 1.628 | 2.047 | 2.987 | $10^4$ |
| Forest Fires | 15.2222 | 15.8722 | 16.4122 | 17.6722 | $10^1$ |
| FPS Benchmark | 0.216 | 0.946 | 1.542 | 2.623 | $10^1$ |
| Geographical Origin of Music | 1.449 | 2.169 | 2.898 | 3.749 | $10^1$ |
| Grid Stability | 0.056 | 0.742 | 1.265 | 2.441 | $10^{-2}$ |
| Health Insurance | 1.4081 | 2.1831 | 2.6581 | 3.8081 | $10^1$ |
| Kin8nm | 0.067 | 0.821 | 1.379 | 2.495 | $10^{-1}$ |
| Kings County | 1.0116 | 1.7886 | 2.2626 | 3.3826 | $10^5$ |
| Miami Housing | 0.271 | 1.008 | 1.613 | 2.724 | $10^5$ |
| Moneyball | 1.828 | 2.528 | 3.143 | 4.328 | $10^1$ |
| Naval Propulsion Plant | 0.039 | 0.580 | 1.128 | 2.349 | $10^{-2}$ |
| Physiochemical Protein | 3.189 | 3.889 | 4.509 | 5.609 | $10^0$ |
| Pumadyn32nh | 2.030 | 2.697 | 3.287 | 4.455 | $10^{-2}$ |
| QSAR Fish Toxicity | 0.684 | 1.398 | 2.066 | 3.084 | $10^0$ |
| Red Wine | 3.619 | 4.355 | 4.945 | 6.089 | $10^{-1}$ |
| Sarcos | 0.171 | 0.861 | 1.472 | 2.605 | $10^1$ |
| Socmob | 1.012 | 1.787 | 2.422 | 3.512 | $10^1$ |
| Solar Flare | 1.016 | 1.753 | 2.386 | 3.511 | $10^{-1}$ |
| Space GA | 0.932 | 1.682 | 2.293 | 3.405 | $10^{-1}$ |
| Student Performance (POR) | 0.976 | 1.642 | 2.261 | 3.339 | $10^0$ |
| Superconductivity | 0.891 | 1.598 | 2.233 | 3.356 | $10^1$ |
| Video Transcoding | 0.056 | 0.775 | 1.302 | 2.319 | $10^1$ |
| Wave Energy | 0.2723 | 1.0123 | 1.5823 | 2.7423 | $10^4$ |
| White Wine | 0.647 | 1.404 | 2.037 | 3.082 | $10^{-1}$ |