# OpenReview forum: "Additive Coupling of Liquid Neural Networks and Modern Hopfield Layer for Regression"
_ICLR.cc/2026/Conference — ICLR 2026 Conference Withdrawn Submission_

### Official Review · Reviewer_eS9H · 2025-10-31

**Soundness:** 3
**Presentation:** 2
**Contribution:** 3
**Rating:** 4
**Confidence:** 4

**Summary:**

This paper proposes a hybrid regression framework that combines the adaptive dynamics of Liquid Time-Constant (LTC) networks with the associative memory of Modern Hopfield Networks (MHN). The LTC component models input-dependent continuous-time dynamics, allowing each neuron to adapt its response to varying inputs, while the MHN compensates for LTC’s lack of an explicit memory mechanism by retrieving patterns from a set of stored prototypes. The two components are integrated through an additive coupling scheme that balances dynamic evolution and memory recall, followed by a lightweight MLP regression head. The authors evaluate the approach on the CTR23 benchmark, which includes 34 tabular regression datasets, and report consistent performance gains over both classical (XGBoost and Random Forest) and neural baselines (vanilla LTC).

**Strengths:**

This paper provides an extensive and well-structured experimental evaluation, demonstrating that the proposed model improves performance on 29 out of the 34 datasets in the CTR23 benchmark. The benchmark itself covers a diverse range of regression tasks, making the evaluation comprehensive. The authors include clear and insightful visualizations, particularly the parity and loss-landscape plots. that effectively illustrate both the strengths and the limitations of their method. Their inclusion of less favorable cases, such as on the Wave Energy dataset, strengthens the credibility of the results by avoiding selective reporting. Finally, the paper’s discussion of limitations is particularly commendable, showing that the authors critically assessed the behavior of their model and acknowledged scenarios where it may not perform as well.

**Weaknesses:**

While the paper presents a coherent hybrid framework, the overall novelty is somewhat limited, as it primarily combines two existing mechanisms, Liquid Time-Constant networks and Modern Hopfield Networks, in a straightforward additive manner. The methodological contribution lies more in the integration than in introducing new concepts. Empirically, the performance gains are modest, with improvements in RMSE that, while consistent, are relatively small across most datasets. The paper would benefit from a more thorough hyperparameter sensitivity analysis, especially regarding key design choices such as the Hopfield memory size (set to 16) and the retrieval scaling factor (β = 0.25). It remains unclear how these values were selected or how they affect performance and stability. Additionally, there is no discussion of computational cost or efficiency, which is important given the added memory and retrieval operations of the Hopfield component. Comparing training and inference times against baselines would have provided a more complete picture of the trade-offs. Finally, there are minor writing and formatting inconsistencies, including grammatical errors, such as in the second paragraph of Section 4.2, and occasional awkward phrasing, which slightly detract from the overall readability of the paper.

**Questions:**

1.	What is the common practice for setting hyperparameters for CTR evaluations?
2.	For clarity and consistency, it would be helpful to format Table 2 in the same way as Table 1, for instance, by bolding the best RMSE and underlining the second-best result.

---

> ### Author Response · Authors · 2025-11-20
>
> *Summary*
>
> We thank all reviewers for their careful reading and constructive feedback. We revised the manuscript to improve clarity, strengthen empirical support, and address missing details. Specifically, we corrected figure scales/captions, expanded Lemma 4 with theoretical variance analysis and dataset-level gradient statistics (Appendix A4), added sensitivity evaluations for memory size and scaling (Appendix B3–B4), and included a computational-cost summary (Appendix B2).
>
> We thank Reviewer `eS9H` for the positive remarks on our paper. We address the concerns below.
>
> 1. **Clarification on Novelty (W1).**
>
>     We acknowledge the reviewers’ observation that our architecture combines two existing components—Liquid Time-Constant (LTC) dynamics and Modern Hopfield Network (MHN) retrieval—through a simple additive fusion. The contribution of this work, however, does not lie in introducing a new architectural primitive, but in establishing a *theoretically grounded and empirically validated interaction* between continuous-time liquid dynamics and associative-memory retrieval for tabular regression.
>
>     LTC networks are well known for modeling adaptive continuous-time dynamics, yet they evolve a single hidden state that must simultaneously encode fast local variations and slower global structure. This makes them prone to hidden-state overwriting: rapidly changing inputs continually update the state, causing the model to lose coarse relationships and long-range dependencies that are important in heterogeneous, nonlinear tabular datasets.
>
>     To address this intrinsic weakness, we introduce a lightweight coupling mechanism in which an MHN retrieves a stable memory-consistent correction that is injected back into the liquid state at every step. Although the coupling is additive, it creates a new computational behavior: the liquid trajectory is repeatedly steered toward associative attractors while preserving its continuous-time dynamics. We also show that this interaction remains bounded and contractive (Lemmas 1–3), and we further demonstrate smoother and lower-variance gradients (Lemma 4) supported by empirical statistics across CTR23 datasets.
>
>     Thus, the novelty of our contribution lies not in architectural complexity but in showing that memory-augmented continuous-time flow models can be made stable, interpretable, and consistently more accurate through a theoretically principled coupling. The extensive evaluation—full CTR23 suite, ablations, loss-landscape analysis, and sensitivity tests—provides comprehensive evidence that this integration meaningfully improves regression performance with minimal overhead.
>
> 2. **Hyperparameter sensitivity and Computational Cost (Q1)**
>
>     We thank the reviewer for highlighting the need to better characterize the influence of the Hopfield memory size and retrieval scaling factor. In the revised manuscript, we have added a dedicated sensitivity analysis in Appendix B3 and B4, where we systematically vary the scaling factor $\beta \in \{0.25, 0.5 0.1, 4.0, 8.0\}$ and memory size in $\{16, 32, 64, 128\}$. Moreover, we have also added another section discussing model size and computational cost in appendix B2.
>
> 3. **Formatting (Q2)**
>
>     We have revised manuscript for clarity, corrected minor phrasing inconsistencies, and reformatted Table 2 to match Table 1, including bolding and underlining conventions.

---

### Official Review · Reviewer_nVET · 2025-11-04

**Soundness:** 3
**Presentation:** 4
**Contribution:** 3
**Rating:** 8
**Confidence:** 2

**Summary:**

This paper proposes a hybrid architecture for structurally complex tabular regression data. It uses Liquid Neural Networks (LNNs) to process continuous-state data adaptively. To address the fact that LNNs are inherently local in time and feature space, the authors apply Modern Hopfield Networks (MHNs) for memory retrieval. They then use simple additive coupling to combine these two parts. Evaluation on OpenML-CTR23 shows that each part of the architecture is necessary and effective, as claimed.

**Strengths:**

1.	The idea is simple and direct: use an LNN for continuous data processing and add an MHN to provide memory retrieval.
2.	The paper presents lemmas establishing boundedness, implying smooth gradients and overall system stability.
3.	The ablation study on the additive LNN+MHN design shows each component works as intended; the MHN indeed supplies memory and is essential.
4.	On OpenML-CTR23, the architecture outperforms competing methods.
5.	The writing is clear and easy to follow.

**Weaknesses:**

1.	Comparisons to related OpenML-CTR23 work. As noted in the Introduction, this paper also reports results on OpenML-CTR23. Please add direct comparisons to the methods cited there (e.g., ‘Accurate predictions on small data with a tabular foundation model’) or briefly justify why such comparisons are not appropriate.
2.	Compute/efficiency reporting. Please add a small table like params or FLOPs.
3.	In Figure 2, the legend refers to orange tracks, but none are visible.



Noah Hollmann, Samuel Muller, Lennart Purucker, Arjun Krishnakumar, Max K ¨ orfer, Shi Bin Hoo, ¨
Robin Tibor Schirrmeister, and Frank Hutter. Accurate predictions on small data with a tabular
foundation model. Nature, 637(8045):319–326, Jan 2025. ISSN 1476-4687. doi: 10.1038/
s41586-024-08328-6. URL https://doi.org/10.1038/s41586-024-08328-6.

**Questions:**

1.	Is there a formal reproducibility statement? Any plan to release code and model checkpoints?
2.	Please include an LLM usage statement (per venue policy).
3.	Additional comments: I am not familiar with this topic. The paper is clearly written and, on that basis, I am inclined to give a high rating, but with low confidence. My rating may change during the rebuttal stage.

---

> ### Author Response · Authors · 2025-11-20
>
> *Summary*
>
> We thank all reviewers for their careful reading and constructive feedback. We revised the manuscript to improve clarity, strengthen empirical support, and address missing details. Specifically, we corrected figure scales/captions, expanded Lemma 4 with theoretical variance analysis and dataset-level gradient statistics (Appendix A4), added sensitivity evaluations for memory size and scaling (Appendix B3–B4), and included a computational-cost summary (Appendix B2).
>
>
> We thank Reviewer `nVET` for the positive evaluation of our paper and constructive comments. We address the concerns below.
>
> **1. Comparisons to related OpenML-CTR23 work (W1).**
>
> The cited tabular foundation model requires dataset-specific pretraining, feature tokenization, and large-scale augmentation pipelines that are not part of the CTR23 evaluation protocol. CTR23 was designed to compare models that operate natively on tabular vectors under fixed train/test splits, which is why the benchmark includes only XGBoost, Random Forest, GAM, and MLP-based baselines. Introducing a heavy pretraining-based model would create methodological asymmetry.
>
>
> **2. Computational efficiency reporting (W2).**
>
> We have added modelsize and computational cost subsection to the appendix B2. We have also included effect of hopfield scaling ($\beta$) and MHN size on proposed method in sections B3 and B4.
>
> **3. Issues in Figures 2 (W3).**
>
> We thank the reviewer for pointing out these inconsistencies. The orange-track caption error in Figure 2 is now corrected.
>
> The code will be released upon publication.

---

### Official Review · Reviewer_GaVQ · 2025-11-07

**Soundness:** 3
**Presentation:** 3
**Contribution:** 2
**Rating:** 4
**Confidence:** 3

**Summary:**

This paper proposes a hybrid regression model that couples Liquid Time-Constant (LTC) networks with Modern Hopfield Networks (MHNs) through a simple additive fusion mechanism. Tested on the OpenML-CTR23 benchmark of 34 regression datasets, the proposed model achieves consistent gains—averaging a 10.4% reduction in RMSE over baselines, including XGBoost and vanilla LNNs. Theoretical proofs demonstrate bounded, stable dynamics and smoother gradient behavior.

**Strengths:**

1. Lemmas 1–3 are clearly stated, which makes it easy for me to follow, even for readers who are not specialists in this field.
2. The experiments on the CTR23 datasets demonstrate the robustness of the proposed method and show that it outperforms baselines in most scenarios.
3. The proposed model inherits the strengths of LTC in terms of local-region regression and stability.

**Weaknesses:**

1. In Lemma 4, the authors assert that the linear combination of gradients can **reduce variance and aid convergence**. However, they should provide either theoretical proof or empirical evidence to support this claim.
2. The authors do not report standard deviations ($\pm$ std) or confidence intervals for their experimental results.
3. A key limitation of continuous-time regression models is **temporal generalization**—i.e., the model is trained on a training set whose time range does not overlap with that of the test/validation set. Otherwise, the task is similar to the standard function fitting on scatter plots. In the conclusion section, the author said they overcame the accumulation of errors over long horizons. Did they refer to the temporal generalization? However, I cannot find a description of how to split the training and test sets to ensure that the time ranges are non-overlapping.

**Questions:**

1. The authors compare their proposed method with traditional machine-learning regressors (XGBoost, Random Forest, Ridge Regression, etc.). Have you considered comparisons with state-of-the-art neural network–based regression models?

2. When comparing the proposed method with other models, you should evaluate not only the MSE metric, but also report the computational complexity, runtime (wall-clock time), and possibly parameter efficiency.

3. In Figure 2, I cannot find the orange tracks mentioned in the caption. Is this a typo or a missing step to color the curves?

4. In Figure 3, the authors state that the proposed model exhibits broader and smoother basins on the Brazilian Houses dataset. However, the y-axis units differ: the LTC plot uses a scale of 1, while the proposed model’s plot uses $10^4$. Does this mean that LTC is actually smoother in this visualization?

5. The experiments focus on low-dimensional real-world datasets. Have you tested the model on multi-frequency or high-dimensional synthetic examples, such as $y = \sin(1/t)$ or $y = \sum_{i=1}^{100} \cos(t_i)+\sin(t_{101-i})$? It is a challenge for most neural network-based models.

6. The author can explore the temporal generalization of their proposed model.

---

> ### Author Response · Authors · 2025-11-20
>
> *Summary*
>
> We thank all reviewers for their careful reading and constructive feedback. We revised the manuscript to improve clarity, strengthen empirical support, and address missing details. Specifically, we corrected figure scales/captions, expanded Lemma 4 with theoretical variance analysis and dataset-level gradient statistics (Appendix A4), added sensitivity evaluations for memory size and scaling (Appendix B3–B4), and included a computational-cost summary (Appendix B2).
>
>
> We thank reviewer `GaVQ` for taking their time in reviewing our paper. Below we provide clarifications to the expressed concerns -
>
> 1. **Lemma 4 (W1, W2).**
>
>     We appreciate this observation and have substantially expanded Lemma 4 in the appendix section A4 of the revised manuscript. Specifically, we now provide-
>
>     - A full variance-decomposition proof showing that$Var(g_z) < max\{(Var(g_x), Var(g_r)) \}$, under the mild and empirically supported assumption of weak covariance between the two gradient paths; and
>     - empirical gradient statistics in form of mean and standard deviation aggregated across all 34 CTR23 datasets, confirming that the coupled model consistently reduces gradient variance. These additions resolve the concern regarding convergence-related justification.
>
> 2. **Temporal generalization (W3, Q6).**
>
>     We recognize that our earlier phrasing may have caused confusion. This paper does not address temporal generalization, as CTR23 contains no temporal axis. We revised the manuscript to remove temporal language and now explicitly emphasize feature-space drift and long-range regression behavior. For context, the original LTC paper already demonstrates temporal extrapolation capability, and extending memory-augmented LTCs to temporal settings is now highlighted as a promising direction for future work.
>
> 3. **Comparison with SOTA neural regressors (Q1).**
>
>     CTR23 is a fully tabular benchmark with no sequential, spatial, or token structure. Modern SOTA neural regressors such as Transformers, S4/SSMs, and similar architectures assume ordered or tokenized inputs and cannot operate natively on unordered feature vectors without imposing artificial structure, which would violate CTR23’s evaluation protocol. For this reason, CTR23’s official baselines include only tabular-appropriate models. We follow this standard to ensure a fair and meaningful comparison.
>
> 4. **Computational efficiency reporting (Q2).**
>
>     Following the comments of Reviewers `GaVQ` and `nVET` , we have added model size, computational cost, and an analysis of the effects of Hopfield memory size and scaling factor $\beta$ in the appendix sections B2, B3 and B4.
>
> 5. **Issues in Figures 2 and 3 (Q3 and Q4).**
>
>     We thank the reviewer for pointing out these inconsistencies. The orange-track caption error in Figure 2 is now corrected. For Figure 3, the scientific-notation scale was unintentionally suppressed during rendering. We have updated the plots to display the correct z-axis scale, resolving the apparent mismatch.
>
> 6. **Synthetic high-dimensional signals (Q5).**
>
>     The reviewer’s suggested functions lie in the frequency-domain function-approximation setting. CTR23, however, is a tabular regression suite with fixed spatial (feature-based) structure. Our method is designed specifically for stable regression under heterogeneous feature interactions rather than modeling oscillatory spectral patterns. Due to this domain mismatch, we did not include frequency-domain synthetic benchmarks. We now note that extending memory-augmented LTCs to high-frequency temporal signals is an interesting direction for future work.

---

> > ### Comment · Reviewer_GaVQ · 2025-11-24
> >
> > **W1**: Do you assume the covariance $\text{Cov}(g_x, g_r)$ is close to zero? Thus,
> > $\text{Var}(g_z) = \text{Var}(\alpha g_x + \delta g_r) = \alpha^2 \text{Var}(g_x) + \delta^2 \text{Var}(g_r) \le (\alpha^2+ \delta^2) \max(\text{Var}(g_x), \text{Var}(g_r)) \le \max(\text{Var}(g_x), \text{Var}(g_r)).$
> >
> > **Q2**: Thank you for providing additional experiments, which show how the hyperparameters $\beta$ and $M$ affect the model size and RMSE. However, I am confused about the experimental results in Table 7. You present results on 34 datasets, and in each, when you **scale up** the model, the RMSE also **increases**, which contradicts my experience. Additionally, I believe that reporting wall-clock time remains necessary when evaluating runtime efficiency.

---

> > > ### Author Response · Authors · 2025-11-26
> > >
> > > We thank the reviewer for the constructive follow-up comments. We address each point below.
> > >
> > > 1.	Our proof of Lemma 4 in appendix section A.4 relies on a weak-covariance assumption that $\mathrm{Cov}(g_x, g_r)$ is close to zero.
> > > To validate this empirically, we measured the covariance across all 34 CTR23 datasets, sampling gradients from matched checkpoints. The observed covariances were extremely small (ranging between $10^{-3}$ and $10^{-11}$). This confirms that the cross-term $2\alpha \delta Cov(g_x, g_r)$ is negligible in practice. We have clarified the assumption and the empirical validation in the revised manuscript.
> > >
> > > 2. Modern Hopfield Networks (MHNs) theoretically provide an exponential storage capacity that scales with the dimensionality of the memory vectors (Ramsauer et al., Hopfield Networks is All You Need, NeurIPS 2021). Increasing the memory size $M$ therefore enlarges the space of patterns that could be stored. However, the same paper shows that when the number of actual stored patterns is small relative to the memory dimension, the energy landscape becomes populated with metastable (spurious) attractors. In such cases, the retrieval dynamics may drift toward these spurious attractors instead of converging to a meaningful stored prototype. This effect becomes more pronounced as $M$ grows large while the underlying dataset provides only a limited number of associative patterns.
> > >
> > >    This explains our empirical findings: the CTR23 datasets contain only a small amount of associative structure, so a large $M$ effectively overparameterizes the memory space. As a result, the retrieval distribution becomes noisier, and the liquid state is more easily attracted to suboptimal metastable states—leading to a degradation in performance (higher RMSE) as $M$ increases. We have clarified this point in the revised manuscript and now note it explicitly as a limitation.
> > >
> > > 3. Regarding runtime efficiency, we reported parameter counts, and FLOPs as inference costs. Wall-clock timing varies substantially across hardware and software stacks, so we opted for FLOPs-based reporting for reproducibility. We agree with the reviewer that a deeper exploration of runtime–accuracy trade-offs is valuable and we aim to pursue it in future work.
> > >
> > > We hope that the clarifications and improvements address the reviewer’s concerns, and we respectfully submit that the work provides a meaningful, simple yet effective contribution with theoretical validation that will generate good discussion at the ICLR conference.

---

### Author Response · Authors · 2025-12-02

We sincerely thank all reviewers for their detailed and thoughtful evaluations. We took every concern seriously, and the revision includes substantial updates across analysis, theory, and empirical support. The manuscript now provides clearer framing, stronger justification, and more transparent reporting. Below we summarize the main issues identified in the reviews and the concrete steps taken to resolve them in the revised version.

---

## **1. Variance reporting, compute cost, and sensitivity analysis** (`GaVQ, nVET, eS9H`)

We acknowledge that the initial submission did not adequately report variance, computational cost, or the sensitivity of key hyperparameters. The revised manuscript addresses these gaps thoroughly:

- **Appendix B2** now includes detailed model size and FLOPs, making computational cost explicit.

- **Appendices B3–B4** provide a systematic sensitivity study of the Hopfield memory size $M$ and scaling factor $\beta$.

- We expanded the discussion of MHN behavior when $M$ becomes large, explaining retrieval degradation via metastable attractors.

- New empirical results supporting these analyses are now reported in **Tables 5, 6, 7, and 8**.


These additions make the method’s computational and behavioral characteristics transparent across a wide range of configurations.

---

## **2. Missing theoretical justification for variance reduction in Lemma 4** (`GaVQ`)

We agree that the original argument for Lemma 4 was incomplete. In the revision, we substantially strengthened this sub-section:

- **Appendix A.4** now presents a full variance-decomposition proof.

- We measured gradient covariances across all 34 CTR23 datasets and found extremely small values ($10^{−3}$ to $10^{−11}$), confirming that the cross-term is negligible in practice.

- **Table 3** reports dataset-level gradient mean/std, empirically validating the smoothing effect predicted by the theory.


These changes provide a complete and well-supported justification for the reduction of variance in our proposed method.

---

## **3. Misleading implication of temporal generalization** (`GaVQ`)

We acknowledge that the initial phrasing could be interpreted as claiming temporal extrapolation, which CTR23 does not evaluate. To correct this:

- All temporal-generalization language has been removed.

- The discussion now clearly focuses on feature-space stability.

- Temporal generalization is listed explicitly as future work, separate from the present contributions.


---

## **4. Comparison to tabular foundation models** (`nVET`)

The reviewer raised an important question regarding comparisons to recent tabular foundation models. We clarified this point directly:

- Such models rely on pretraining, tokenization, and augmentation pipelines that do not align with CTR23’s controlled tabular evaluation protocol. Including them would introduce methodological asymmetry. We explicitly mark comparison to these models as future work.

- Although broader comparisons to tabular foundation models would be valuable, our current evaluation already provides strong evidence. The results across all 34 CTR23 datasets, supported by ablations and sensitivity studies, show clear and consistent benefits of our proposed method.


---

## **5. Figure corrections and general clarity** (`all reviewers`)

We made a careful pass through the entire manuscript to improve clarity:

- Corrected caption errors and restored missing axis scaling.

- Fixed notation issues and formatting inconsistencies.

- Improved readability in multiple sections.


---

The revised paper now presents a simple and transparent coupling of continuous-time dynamics with associative recall, supported by theory and extensive empirical evidence. The evaluation covers all 34 CTR23 datasets, includes negative and borderline cases, and clearly states the method’s limitations. We believe these revisions fully address the reviewers’ central concerns and strengthen the work substantially.

We remain hopeful that these improvements will be reflected in the final assessment, and we believe the paper now stands as a solid and well-supported contribution for presentation and discussion at **ICLR 2026**.

---

### Note · Authors · 2026-03-19

I have read and agree with the venue's withdrawal policy on behalf of myself and my co-authors.

---

### Meta-Review · Area_Chair_gJ2y · 2026-01-07

**Summary:**

Reviewer GaVQ: Lemmas 1–3 are clearly stated, which makes it easy for me to follow, even for readers who are not specialists in this field. The experiments on the CTR23 datasets demonstrate the robustness of the proposed method and show that it outperforms baselines in most scenarios. The proposed model inherits the strengths of LTC in terms of local-region regression and stability. However, the reviewer still has some concerns on the weaknesses about lack of theoretical proof or empirical evidence, key limitation of  temporal generalization.


 Reviewer nVET: The idea is simple and direct: use an LNN for continuous data processing and add an MHN to provide memory retrieval. The paper presents lemmas establishing boundedness, implying smooth gradients and overall system stability. The ablation study on the additive LNN+MHN design shows each component works as intended; the MHN indeed supplies memory and is essential. However, the reviewer still has some concerns on the weaknesses about lack of more comparisons, lack of efficiency reporting, unclear figure details.

Reviewer eS9H: This paper provides an extensive and well-structured experimental evaluation, demonstrating that the proposed model improves performance on 29 out of the 34 datasets in the CTR23 benchmark. The benchmark itself covers a diverse range of regression tasks, making the evaluation comprehensive. The authors include clear and insightful visualizations. However, the reviewer still has some concerns on the weaknesses about limited novelty, lack of hyperparameter sensitivity analysis, lack of discussion of computational cost or efficiency.

**Reviewer Concerns:**

After carefully evaluating the rebuttals, I think the reviews from the Reviewer nVET were addressed from the response.
For the remaining reviewer concerns, they are all not fully addressed.

**Reviewer Scores:**

For the  Reviewer  nVET, I think the reviewer may increase the rating or keep the rating unchanged based on the response.
For the remaining Reviewers, I think the reviewer may keep the rating unchanged based on the response.

---

### Decision · Program_Chairs · 2026-01-26

Reject